# Aptamers: Novel Therapeutics and Potential Role in Neuro-Oncology

**DOI:** 10.3390/cancers12102889

**Published:** 2020-10-09

**Authors:** Paola Amero, Soumen Khatua, Cristian Rodriguez-Aguayo, Gabriel Lopez-Berestein

**Affiliations:** 1Department of Experimental Therapeutics, The University of Texas MD Anderson Cancer Center, Houston, TX 77054, USA; PAmero@mdanderson.org; 2Division of Pediatrics, The University of Texas MD Anderson Cancer Center, Houston, TX 77030, USA; SKhatua@mdanderson.org; 3Center for RNA Interference and Non-Coding RNA, The University of Texas MD Anderson Cancer Center, Houston, TX 77054, USA; 4Department of Cancer Biology, The University of Texas MD Anderson Cancer Center, Houston, TX 77030, USA

**Keywords:** aptamers, therapeutic, diagnostic, blood–brain barrier, pediatric, neuro-oncology

## Abstract

**Simple Summary:**

Despite the remarkable effort of researchers to find more effective treatments for pediatric brain tumors, the prognosis continues to be poor. Forty percent of pediatric patients develop treatment resistance and relapse, and the morbidities and long-term side effects of systemic therapy remain concerning. Significant advances have been made by next-generation genomic profiling. Novel oncogenic drivers have been identified as potential targets for the treatment of pediatric brain tumors. Aptamers, which are synthetic single-strand oligonucleotides, specifically target, bind with high affinity, internalize, and deliver a wide range of therapeutic moieties inside the cells. Although several aptamers have been tested in preclinical and clinical studies for adult glioblastoma, the use of aptamers in pediatric neuro-oncology remains unexplored. Increased knowledge regarding the molecular mechanisms of the pathogenesis of pediatric brain tumors, as well as selection of novel aptamers and/or adaptation of aptamers currently used in adult glioblastoma, might open a novel research field.

**Abstract:**

A relatively new paradigm in cancer therapeutics is the use of cancer cell–specific aptamers, both as therapeutic agents and for targeted delivery of anticancer drugs. After the first therapeutic aptamer was described nearly 25 years ago, and the subsequent first aptamer drug approved, many efforts have been made to translate preclinical research into clinical oncology settings. Studies of aptamer-based technology have unveiled the vast potential of aptamers in therapeutic and diagnostic applications. Among pediatric solid cancers, brain tumors are the leading cause of death. Although a few aptamer-related translational studies have been performed in adult glioblastoma, the use of aptamers in pediatric neuro-oncology remains unexplored. This review will discuss the biology of aptamers, including mechanisms of targeting cell surface proteins, various modifications of aptamer structure to enhance therapeutic efficacy, the current state and challenges of aptamer use in neuro-oncology, and the potential therapeutic role of aptamers in pediatric brain tumors.

## 1. Introduction

Theoretically, nucleic acids have a multifunctional nature. Synthetic single-strand nucleic acid aptamers are able to form complex shapes acting as a scaffold for molecular interactions, with high specificity and affinity. This suggests that nucleic acids not only store hereditary information but also may have other functions [1]. Aptamers are synthetic single-strand oligonucleotides that can bind and regulate the activity of proteins based on their tertiary structural interactions. They have the potential to specifically target, bind with high affinity, internalize, and deliver a wide range of therapeutic moieties inside the cells. A variety of chemical modifications and polyethylene glycol (PEG) conjugation confer a pharmacokinetic profile that can overcome metabolic instability. PEGylation reduces renal filtration and improves aptamer bioavailability. Toxicology studies have demonstrated that aptamers are non-immunogenic and cannot activate the immune system, although PEG can lead to hypersensitivity reactions owing to anti-PEG antibodies. Generally, aptamers appear safe; no off-target pharmacologic effects have been reported, and the adverse effects depend on the target [2].

Since aptamer-based technology was introduced two decades ago, its potential in therapeutic and diagnostic fields has been very evident [3]. The discovery that nucleic acids modulate the activity of target proteins acting as ligands goes back to studies of the biology of human immunodeficiency virus. These studies showed that specific structured RNA sequences could either facilitate the replication or reduce the activity of human immunodeficiency virus [4,5,6]. Recently, interest in aptamers as therapeutic or diagnostic tools has significantly increased. The oligonucleotide composition of aptamers presents several advantages for clinical application, such as low systemic toxicity [7]. Adverse effects associated with aptamer therapy are rare, but when they are present, they may arise from polyanionic effects, tissue accumulation, and/or intense chemical modifications [8]. In therapeutic applications, aptamers can serve in three different roles: as antagonists, blocking interactions with targets associated with various diseases through protein-protein or ligand-receptor interactions; as agonists, activating the function of target molecules; and as carriers of therapeutic moieties, delivering therapeutic agents to target cells or tissues [8,9]. Because aptamer-target interactions are similar to antibody-antigen complexes, aptamers have been considered “chemical antibodies” [8,10]. Therapeutic antibodies have been used to improve the lack of specificity seen with chemotherapies, leading to the development of novel drugs. Although antibodies are a potent therapeutic tool, their use for the treatment of brain diseases, including neurodegenerative diseases and specifically brain cancers, has been limited, particularly due to the blood–brain barrier (BBB), which makes brain tissue very difficult to access by traditional antibody strategies. Along with overcoming these limitations, features such as low immunogenicity, high stability, easy tissue penetration, and accessibility to chemical modifications make aptamers preferred over conventional therapeutic antibodies [8,11].

Central nervous system tumors are the second most common type of cancer and the most common solid tumor among children in Europe and the United States. Current treatment strategies are maximal safe surgical resection, radiotherapy, and chemotherapy, all of which are associated with significant adverse effects and long-term sequelae. Novel targeted therapeutic strategies are needed in pediatric neuro-oncology to improve survival and reduce adverse effects [12,13].

Imaging, early detection, and identification of the cancer stage are key elements to developing new therapeutic and diagnostic approaches. Aptamer-based technologies provide great opportunities, combining diagnostic and therapeutic applications [14]. Given their accessible chemical modifications, aptamers can be physically or chemically conjugated to a wide range of probes and therapeutic agents [15]. Many studies have reported these advances, highlighting the potential future role of aptamers as promising agents for imaging and detection in cancer therapy.

In this review, we discuss the current state and challenges of aptamer use in neuro-oncology with a focus on pediatric neuro-oncology, a field in which aptamers have not been explored.

## 2. Aptamers

Given the unique repertoire of special characteristics such as target selectivity, high potential affinity, simplicity of modification, scalability, flexibility, low immunogenicity, and wide drug delivery platform, aptamers have emerged as an exciting and promising novel diagnostic and therapeutic tool for brain tumors [16]. Here we summarize the latest systematic evolution of ligands by exponential enrichment (SELEX) methods, post-selection modifications, and the use of aptamers as diagnostic and therapeutic agents.

### 2.1. Latest Advanced SELEX Methods

Aptamers are designed and selected through repeated rounds of selection (i.e., SELEX). In 1990, two independent research groups developed the SELEX method [4,5], and several studies have subsequently used this method. Tuerk and Gold were the first to select an RNA sequence (from a DNA library) that could recognize T4 DNA polymerase in vitro [5]. Ellington and Szostak were the first to coin the term “aptamer” and identify RNA populations (from a large library of random RNA sequences) that could bind various organic dyes [4,17]. SELEX is now a well-established method.

Since its establishment, new and more effective procedures have been developed to increase the efficiency of aptamer selection [18]. For example, most in vitro SELEX methods require solid supports to immobilize target molecules [19]. These solid supports are suitable for several targets because they come in a variety of forms, such as resin, magnetic beads, and several tags (biotin, histidine, glutathione S-transferase, and streptavidin). In addition, methods such as quantitative screening particle display, capillary electrophoresis SELEX, atomic force microscopy, flow cytometry, microfluidics, sol-gel–based microfluidics, biacore surface plasmon resonance, and microarray-based SELEX have been successfully incorporated into the conventional SELEX to improve its ability to identify high-affinity aptamers, as well as its general efficiency [18,20].

In contrast, non-SELEX aptamer selection and high-fidelity methods miss the important amplification steps needed to decrease the errors related to insertion of the wrong nucleotide or too many or too few nucleotides into a sequence produced by the polymerase chain reaction procedure [21,22]. The major advantages associated with the use of conventional SELEX are reduced time needed to complete the process and increased affinity for the target [23,24].

Whereas in vitro SELEX can recognize only known target proteins, cell-SELEX [25], slide-based SELEX [26], and in vivo SELEX [27] directly target unknown target molecules expressed on whole cells. In addition, Morph-X-Select technology, developed by Wang et al., can simultaneously select aptamers and identify unknown biomarkers using pathologic tissue sections [28].

One variant of SELEX that can target not only tumoral cells but also morphologically or structurally complex healthy tissues is in vivo SELEX. Cheng et al. [29] identified brain-penetrating aptamers using in vivo SELEX. In this method as initially described, in wild-type mice, RNA aptamers were identified that could cross the BBB and reach the central nervous system after peripheral injection. The initial 2′-fluoropyrimidine–modified RNA library was administered in mice via tail vein injection and recovered by the brain. Twenty-two rounds of selection were performed. Three sequences were identified, named A02, A09, and A15. The homing potential of these aptamers was tested using a reverse screening assay. A15, A09, A02, and scrambled aptamer were mixed in equal molar ratios and administered to mice by tail vein injection. The results showed that 45% of the clones extracted from the brain were represented by A15 sequences, 30% by A9 sequences, and 25% by A2 sequences. Nuclease resistance was increased by modifying A15 to incorporate 2′-O-methyl residues. Furthermore, in situ hybridization was performed to assess aptamer brain penetration. Both aptamers, A15 and control aptamers, were injected into mice and specific probes were used that could hybridize the A15 random region. Positive signals for 15 aptamers were revealed in different regions of the brain, including the cortex, hippocampus, cerebellum, and striatum, compared with scramble aptamer [29]. These results suggest that aptamer-based technology is a simple yet powerful tool that holds potential in neuro-oncology.

### 2.2. Structure and Target Association/Binding of Aptamers

Through the complexity of the three-dimensional structure, aptamers are designed to selectively distinguish the target from among different biological structures. Aptamers can bind tightly to a wide spectrum of molecular targets, peptides, proteins, viruses, bacteria, and whole cells [8].

Intramolecular interactions, such as hydrogen bonds, hydrophobic interactions, van der Waals interactions, and aromatic stacking, ensure that the aptamer-target complex functions correctly. Modifications of the binding regions can affect aptamer stability, affinity, and specificity [30]. Three-dimensional folding structure is crucial for aptamer binding and specificity. The binding affinity is measured and given by the equilibrium dissociation constant (KD). Low KD values (from picomolar to nanomolar concentrations) are correlated with high binding affinity for the molecular targets. Specificity indicates the amount of off-target binding, measured by the ratio between the KD of the specific target and the KD of the nonspecific target [7]. Various elements may constitute the aptamer secondary structures, including stem-loop or hairpin, bugle, pseudoknot, DNA G-quadruplex, and kissing hairpin structures, typically depending on the propensity of the structures to form complementary base pairs [8].

Various analytical methods have been used to understand the molecular interactions of aptamer structure and aptamer-target complexes [31]. ClustalW software is one of the most used and cited multiple-sequence alignment tools. However, updated algorithms have been developed to help to identify self-hybridized structures for aptamers and structural patterns between aptamer spatial conformation and protein domains [32]. For example, competition-enhanced ligand screening is a non-evolutionary screening approach in which secondary structure elements and corresponding secondary structure families in aptamers are identified and classified through comparison with a random library [32,33]. Not many analytic tools are available for self-hybridized single-strand DNA analysis. One software program typically used to predict secondary structures for single-strand DNA is mfold modeling software. A few studies have reported a comparative analysis of secondary structure predictions of DNA aptamers, identifying shared binding motifs among different DNA aptamer sequences [34]. AptaTRACE is a platform recently developed for structure motif clustering using secondary structure prediction from SFOLD software. AptaTRACE can predict significant structural enrichment during the SELEX procedure. APTANI software, using RNAsubopt, predicts cluster sequence motifs using substructures, apical loops, bulge loops, and intra-strand loops [35]. Molecular modeling by means of docking and molecular dynamics has turned out to be an indispensable part of the aptamer characterization. However, even after molecular dynamics modeling, molecular docking demonstrates only that the nucleic acid binds well to the target protein. Consequently, docking results should not be over-interpreted before proper experimental validations have been carried out.

### 2.3. Post-SELEX Modifications That Impact the Clinical Translation of Aptamers

Clinical translation of aptamer technologies is limited by inadequate stability of these molecules in biological environments. However, post-selection modifications can be applied to aptamers if the three-dimensional structure and binding properties are preserved [36]. Many strategies have already been developed to improve one or more aptamer properties, such as affinity, nuclease resistance, bioavailability, or thermal stability [37,38].

Nuclease susceptibility represents the biggest challenge in using aptamers in vivo. The fundaments of chiral inversion can be combined with potent iterative selection methods to generate ligands that are not susceptible to degradative enzymes. Naturally, DNAs are synthesized in D-conformation. However, it is possible to generate L-DNA by chemical synthesis, resulting in the mirror image of D-oligonucleotides. The Spiegelmers oligonucleotide backbone is composed of L-ribose linked to phosphodiesters (Figure 1A). This technology can be applied to generate L-aptamers, which are highly resistant to the degradation of nucleases and display stronger affinity to targets. The Spiegelmers aptamer strategy, in which the sugars are enantiomers of the wild type, is based on the concept that nucleases are selective for the D-ribose enantiomer [3,39].

Chemical modifications can be summarized in three strategies. The first is modification of the sugar ring, which involves replacement of 2′ and 4′ positions with 2′-NH2, 2′-F, and 2′-O-CH3, as well as the use of locked nucleic acids, in which the ribose is “locked” with an extra bridge connecting the 2′ oxygen and 4′ carbon, forming an A-form duplex (Figure 1B). These modifications are primarily applied to RNA aptamers conferring high stability and low cytotoxicity [40]. The second strategy is modifications of bases in the 5′ position of pyrimidine and 8′ position of purine. Hydrophobic, hydrophilic, or charged groups can be incorporated in the base to obtain ligands with higher affinity for the target. Study of the functional groups of the target site is needed to select the best groups (Figure 1C) [36]. The third strategy is modifications of the linkage, referring to the backbone and inter-nucleotide linkage [41]. Thiophosphate substitutions in the aptamer backbone are applied for RNA and DNA aptamers in which non-bridging oxygen atoms of the phosphodiester backbones are substituted with one or two sulfur atoms (Figure 1D) [20]. Most of the aptamers that reach clinical trials are thiophosphate-modified, because this modification confers high resistance to nuclease degradation [20,42]. Moreover, for optimization of aptamer pharmacokinetics in vivo, aptamers can be capped at 3′ or 5′ with a wide variety of molecules, such as biotin–streptavidin, inverted thymidine, amine, phosphate, PEG, cholesterol, fatty acids, and proteins, protecting aptamers from exonuclease degradation and prolonging blood circulation by reducing renal filtration [43].

Slow off-rate modified aptamer (SOMAmer) modifications are one of the latest methods of post-selection optimization. SOMAmer modifications functionalize at the 5-position of deoxyuridine residues with moieties (benzyl, 2-napthyl, or 3-indolyl-carboxamide) [44,45]. SOMAmer technology is based on the concept that large and hydrophobic functional groups, miming sidechains of proteins, present better nuclease resistance and selection success rates and improve affinity properties [44,45].

### 2.4. Aptamers for Therapeutic and Diagnostic Applications

To ensure in vivo targeted drug delivery without affecting normal cells, drugs must be released directly to the therapeutic targets, which last for a long time in systemic circulation and are internalized by the target cells. Receptors expressed on the cell surface can be considered targets for specific ligand drugs, such as aptamers [46] (Figure 2A, left panel). Aptamers can be internalized inside the cells mainly through two mechanisms: receptor-mediated endocytosis and macropinocytosis. Negative charge of the phosphate backbone, size, and poor stability can influence the cellular uptake of aptamers [47].

Aptamer-based chimeras represent an important direction for future gene therapies. Aptamers can be physically conjugated by intercalating or covalent linking to novel therapeutics such as noncoding RNA and drugs, serving as targeted signals for the delivery of nanoparticles loaded with active drugs (Figure 2A, right panel) [48]. One of the methods used to link aptamers to drugs is physical intercalation, in which drugs containing an aromatic nucleus, such as doxorubicin, can be loaded to the double strands of DNA [49]. Noncoding RNA such as small interfering RNA, microRNA, and antisense oligonucleotides are structurally akin to aptamers, so their synthesis is enabled with systemic administration. For example, therapeutic RNA can be directly linked with aptamers for cancer therapy. The linkage is constructed through hybridization or a covalent bond [50,51,52]. This is known to contribute to drug delivery owing to advances made in nanotechnology. Aptamer-decorated nanoparticles present several advantages because they allow conjugation of multiple ligands. Different linkage strategies can be applied depending on the aptamer-nanoparticle type. Mainly through oligo-S units (gold nanoparticles), PEG polymer coating [liposome/poly(D,L-lactic-co-glycolic acid)] aptamers can be conjugated to the surface of nanoparticles [53].

Several aptamer-diagnostic approaches have been developed. For example, Jung et al. identified an aptamer-based proteomics method to simultaneously detect seven protein biomarkers (Epidermal growth factor receptor (EGFR), CA6, MMP7, CRP, KIT, C9, and SERPINA3), with high sensitivity and specificity, in cancer patients compared with healthy subjects [54,55,56].

OLIGOBIND is a novel diagnostic approach that uses an aptamer-based enzyme-capture fluorescent assay generated by Sekisui diagnostics. This assay can accurately detect active thrombin levels in blood samples, with a sensitivity in the range of picomolar concentrations. The highly sensitive detection of active thrombin levels, which are a surrogate marker for platelet contamination, enhances the application of this assay [55,56,57].

SOMAscan, based on slow off-rate modified aptamer (SOMAmer), is a biomarker discovery and clinical diagnostic platform created by Larry Gold. This platform can identify with high sensitivity (femtomolar range) more than 1300 proteins simultaneously using a small volume of sample, and it can be used with various types of samples (serum, cerebrospinal fluid, cells, tumor extracts, and synovial fluid; Figure 2B) [55,58].

Accordingly, aptamers seem to be a promising tool for cancer therapeutics, imaging, and diagnostics in biological samples because aptamers are capable of detecting almost any oncologic biomarker, metabolite, or tumoral cell with high affinity and specificity at small target concentrations.

## 3. Aptamers in Neuro-Oncology

Up until 5 years ago, brain tumors were classified by histogenesis, i.e., according to histologic similarities observed under a microscope. However, as a result of some important revelations in molecular biology, as well as RNA, DNA, and methylation profiling data from brain tumors, brain tumors are now classified according to histologic and molecular parameter alterations. This was illustrated in the 2016 World Health Organization (WHO) classification of central nervous system tumors. Glial tumors are now classified as low-grade (WHO grade I and II) or high-grade (WHO grade III and IV), and six molecular subtypes have been identified by methylation assays [59]. Glial tumors, which arise from glial cells, are the most prevalent type of adult brain tumor. There are various important subtypes of gliomas, including diffuse gliomas, non-diffuse gliomas, and ependymomas, classified according to the genotype of the origin glial precursor cell [60]. Conventional treatments include maximal safe surgical resection of the tumor in combination with radiotherapy and chemotherapy. However, most patients with high-grade glioma die within 1 year after diagnosis [61].

One of the major factors undermining the therapeutic efficacy of aptamers in neuro-oncology is poor penetration of the BBB. The BBB is a highly semipermeable neurovascular system responsible for homeostasis and the transport of endogenous and exogenous compounds in the central nervous system. The BBB is composed of tight adherens junctions between adjacent endothelial cells, pericytes, and astrocytic end-feet. Several conditions influence the crossing of drugs through the BBB, including molecular weight (cutoff of 400 to 600 Da for penetration of the BBB), ionization at physiologic pH levels (molecules that have a high electrical charge are slowed), liposolubility (low lipid-soluble molecules do not penetrate into the brain), and protein binding. The physiologic properties of the BBB have thwarted the delivery of anti-neoplastic agents to the central nervous system. To attain therapeutic efficacy, drug delivery must occur in the tumor site at an effective concentration and time. Brain tumors compromise the integrity of the BBB, resulting in BBB disruption. BBB disruption is characterized by non-uniform permeability dependent on tumor type and size and an unselective influx of low and high molecular weight into the brain. The clinical benefit of BBB disruption has not been established for the treatment of less sensitive tumors such as gliomas, but BBB disruption seems to increase survival in patients [62,63]. The delivery of chemotherapeutic agents into the brain may be improved through BBB disruption and across the Ommaya intraventricular reservoir [64,65,66].

Disruption of the BBB and increased leakiness induced by brain tumors are presumed to enhance penetration and facilitate the delivery of drugs to tumor sites (Figure 3A). This is unfortunately not always the case [67]. To date, several aptamers have been identified for therapeutic and diagnostic applications in glioma, including cell surface biomarkers (Table 1 and Table 2) [68]. This novel approach using aptamers has been shown to potentially facilitate BBB penetration and optimize delivery of drugs or other biologic agents to tumors (Figure 3B). Notwithstanding the research efforts showing that disruption of the BBB can increase leakiness and delivery of the chemotherapeutic drugs or aptamer, more research is needed to fully characterize and harness the benefits of this disruption.

### 3.1. Aptamers as Therapeutics in Neuro-Oncology

Many aptamers against oncogenic drivers in glioblastoma (GBM) have been identified [68]. Here, we summarize the latest research related to the use of aptamers alone or combined with other therapeutic moieties in neuro-oncology (Table 1).

Recently, using differential cell-SELEX, Affinito et al. identified A40s, a novel aptamer that can internalize in GBM stem cells and specifically deliver miR-34c and anti-miR10b to the stem cell population. The researchers demonstrated that A40s can cross the BBB to reach the tumor site and selectively bind the EphA2 receptor, thus inhibiting tumor growth and reducing tumor relapse [83,91].

Another group identified an aptamer named GL43.T, which targeted EphB2/3 receptors and inhibited cell vitality, reducing chemotactic serum- and ligand-stimulated cell migration in vitro. However, the effect of GL43.T in animal models has not been reported [80].

Cheng et al. investigated the potential of aptamers to penetrate the BBB in animal models, elegantly demonstrating that aptamers can internalize into endothelial cells through target receptor-mediated transport mechanisms [29].

Oncogenic drivers of GBM have been shown to play a pivotal role in target-therapy development. Epidermal growth factor receptor (EGFR) is upregulated in many cancer types and is associated with cancer progression and poor prognosis. It represents a very interesting target for cancer therapy. Wang et al. described the efficacy of a novel aptamer named CL-4RNV616, which can inhibit the proliferation of U87MG GBM cells by targeting EGFR, and the effect shown by the CL-4RNV616 aptamer warrants further investigation [73].

Another therapeutic strategy used for targeting brain tumors is bifunctional aptamers, in which two sequences are fused together. This strategy can explain how aptamers overcome the BBB, by binding the receptors expressed on epithelial cancer cells and facilitating the internalization of the second aptamer [92]. One example of bifunctional aptamers is the SYL3C aptamer conjugated to the TfR aptamer. The SYL3C aptamer, against Epithelial Cell Adhesion Molecule (EpCAM), was truncated in a sequence of 17-mer and fused to the TfR aptamer, an anti-transferrin DNA aptamer of 14 nucleotides. These aptamers were bound together to build a bifunctional system that could cross the BBB via transcytosis through the TfR aptamer and inhibit metastases through the anti-EpCAM aptamer [92]. A similar strategy was applied by Jun Wei et al. to target infiltrating macrophages. Jun Wei et al. developed the 4-1-BB–Osteopontin (OPN) bispecific aptamer, composed of the OPN-R3 aptamer, against osteopontin (OPN; a glycophosphoprotein upregulated in GBM-infiltrating immune cells such as macrophages and associated with poor survival in GBM patients), to block M0 and M2 macrophage migration, and the A4-1BB aptamer, to promote the survival and expansion of CD8+ T cells. The researchers demonstrated that this bispecific aptamer treatment increased mouse survival in GL261 intracerebral models [81].

Another important technology is the Aptamer Chimera-based technology, which has been extensively used to penetrate the BBB and specifically deliver small interfering RNA and microRNA to GBM cell lines [50,81]. The latest studies in field showed that Gint4.T-STAT3, alone or in combination with GL21.T-10b chimera, inhibited the primary propagation and migration/invasion of GBM stem cells [69]. Larcher et al. developed an oligonucleotide system composed of an enzymatic component to inhibit the expression of miR-21, conjugated to aptamers against transferrin receptor (TfR) to deliver the complex. This antimiRzyme chimera showed efficient inhibition of miR-21 in vitro; however, no animal models have been reported to demonstrate the effectiveness of this catalytic system in vivo [76].

Aptamer-nanoparticle–based nanocarriers are promising delivery systems for anti-glioma therapeutics owing to their ability to cross the BBB and inhibit proliferation, migration, and invasion [73]. Several approaches have been reported. Recently, Monaco et al. described the use of BODIPY@PNPs-Gint4.T nanovector, composed of biodegradable polymeric of poly(lactic-co-glycolic)-block-poly ethylene glycol (PLGA-b-PEG) conjugated to aptamers against platelet-derived growth factor β (PDGFβ), with the potential for basic and translational applications. This novel delivery system can cross the BBB through the enhanced permeability and retention effect in an intracranial murine model, as shown from imaging of whole brains of mice treated with BODIPY@PNPs-Gint4.T [79].

Moreover, Shi et al. developed a three-dimensional tetrahedral framework nucleic acid (tFNA), loaded with anti-PDGFβ aptamer, Gint4.T, and the anti-ssDNA 2 (S2) and -ssDNA 3 (S3) aptamer GMT8, making a complex named Gint4.TtFNA-GMT8 (GTG). The GTG complex has been designed with the aim of facilitating BBB crossing for paclitaxel, a very effective anticancer drug in several cancer types [71]. Aptamer-nanoparticle technology improves the efficacy of drugs currently used in the treatment of GBM. Temozolomide (TMZ), an alkylating agent, can cross the BBB and is widely used for the treatment of GBM, anaplastic astrocytoma, and anaplastic oligoastrocytoma. However, owing to rapid drug resistance, the effectiveness of TMZ is reduced. Fu et al. developed a tFNA nanoparticle loaded with TMZ, carrying AS1411 and GS24 aptamers, to enhance cell apoptosis and autophagy. This novel nanocarrier system was found to be more effective than TMZ alone by overcoming TMZ resistance, killing GBM TMZ-sensitive cells, and activating apoptosis and autophagy pathways. Moreover, in vivo imaging showed that tFNA lasts for 1 hour in the brain vessels, suggesting that tFNA may be a suitable drug delivery system [74].

An aptamer-like peptide strategy for iRNA delivery has been described by Saw et al. This aptamer-like peptide, targeting the extra-domain B (EDB) of fibronectin, was conjugated to the liposome nanoparticle surface by hydrophobic chains of PEG. APT-EDB nanoparticles, systemically administrated, were internalized into target cells and iRNA released into the cytoplasm, leading to inhibition of tumor growth [77].

Aptamers conjugated to cell-penetrating peptides represent another therapeutic strategy to facilitate BBB penetration. Several cell-penetrating peptides have been developed to increase intracellular delivery of molecular cargoes. However, cell-penetrating peptides are limited by lack of selectivity and endosomal entrapment. A novel delivery approach, composed of nanoparticles containing the phage-displayed TGN peptide and the AS1411 aptamer, has been designed to directly penetrate the BBB and target cancer cells without affecting normal brain tissue, reducing toxicity associated with treatments for brain tumors [93,94].

Taken together, these reports indicate that research in aptamer-based technology for neuro-oncology is rapidly progressing. However, the translation of beneficial aptamer therapeutics to the clinical setting is progressing slowly compared with therapeutic antibodies. This might be due to several critical factors, such as lack of knowledge of the nucleic acid chemistry, pharmacokinetics, and biodistribution for aptamers, as well as production costs, competition between aptamers and monoclonal antibody–based drugs, and reluctance to set aside conventional therapies.

### 3.2. Aptamers for Diagnostics and Imaging in Neuro-Oncology

The development of aptamer technology, combined with identification of novel oncogenic biomarkers, may have translational potential for early cancer detection [56]. In the past decade, several aptamer approaches for brain tumor diagnosis and imaging have been reported in the literature (Table 2), and most of these approaches are currently being tested in clinical trials.

Recently, Georges et al. identified a new diagnostic method to identify tumor margins and neoplastic tissue from surgical resection of brain tumors. This method was based on the use of an aptamer labeled with Alexa-488, TD05-488, generating a tissue-specific diagnosis in 20 minutes [82].

Previously, Georges et al. had described another approach based on a conformational FRET-based switchable aptamer. This fluorescent probe quickly identified tumor cells in biopsies from xenograft mouse models. A fluorophore, Alexa-488, and a quencher, Black Hole Quencher, chemically incorporated into an aptamer sequence at the 5′ and 3′ ends, respectively, composed the probe. The quenching properties were tested by flow cytometry. This strategy was based on the concept that when the aptamer is in unbound conformation, the fluorescent signal is minimal, whereas when the aptamer has bound the target, the quencher is released and the fluorescence consistently increased [88].

Magnetic resonance imaging and positron emission tomography represent the standard methods for brain tumor diagnosis [95]. Radiolabeled aptamers are frequently used for positron emission tomography and magnetic resonance imaging analysis in animal models [96,97,98]. Gu et al. described the use of GBI-10, attached to gadolinium-loaded liposomes, to enhance magnetic resonance imaging diagnosis [89]. The use of ^99^mTc radiolabeled-TTA1 and ^188^Re-labeled U2 aptamers for imaging and diagnosis has also been described. These aptamers, systemically injected into tumor-bearing mice, were successfully uptaken into tumors in GBM xenograft animal models and accumulated in areas where their targets were expressed, thus demonstrating that aptamers may be used for GBM imaging and diagnosis [96].

Cell detection though aptamer-based technology has been extensively explored in GBM cells and GBM stem-like cells via flow cytometry, confocal microscopy, and histochemistry for in vitro and in vivo fluorescence imaging [96].

According to confocal microscopy, aptamer internalization and target co-localization have been tested in U87MG cells using H02, a cyanine 5-labeled aptamer against integrin α5β1. Moreover, molecular probe aptamers have been applied for histochemistry and detection of cancer biomarkers in tumor tissues from patient-derived xenografts [84]. Sun et al. described dual-targeting ligand aptamers for therapeutic and diagnostic purposes. This approach combined angiopep-2 to facilitate BBB penetration and A15 aptamer to track CD133+ expression on the surface of glioma stem-like cells. This bifunctional system was assembled on a cationic liposome to deliver anti-neoplastic agents or siRNA. Using in vivo fluorescence imaging, the authors demonstrated that particles covered with bifunctional aptamers more efficiently penetrated the BBB and accumulated in the tumor compared with the particle alone in U251-CD133+ glioma tumor-bearing nude mice [75].

## 4. Potential Role of Aptamers in Pediatric Neuro-Oncology

### 4.1. Pediatric Brain Tumors and Current Treatments

Pediatric brain tumors are the most common solid tumor and the main cause of cancer-related death in children. Medulloblastoma, gliomas, and ependymal tumors constitute the most malignant pediatric brain tumors [99]. Even with the development and clinical use of molecularly targeted agents, the outcome of many patients with malignant brain tumors remains dismal, and relapse is very common.

Among pediatric brain tumors, medulloblastoma has been the most well-studied from a genomic and epigenomic perspective. Medulloblastoma is a malignant embryonal tumor of the developing cerebellum. It has been classified into four distinct molecular subgroups: Wnt, Sonic Hedgehog, group 3, and group 4 [100,101]. In patients with medulloblastoma, death is rarely caused by the primary tumor itself or tumor reappearance at the primary site; instead, death is usually caused by metastasis at the time of recurrence. Although medulloblastoma metastases are detected by imaging in around 35% of patients at the time of initial diagnosis, most patients are later found to have diffuse leptomeningeal metastases. Therefore, irradiation of the whole brain and spinal cord is recommended as the standard of care for prophylaxis against metastases in all patients older than 3 years. However, this procedure is also a major source of morbidity in survivors, causing significant detrimental effects on intelligence and neurologic and endocrine function, as well as the development of secondary radiation-induced neoplasms [102]. Clearly, making the leptomeningeal compartment a point of focus for research is warranted to decrease mortality and morbidity in patients with medulloblastoma.

Pediatric brain tumors are now known to be different from brain cancer in adults. The best example is pediatric high-grade gliomas, which continue to have a poor prognosis, with 2-year survival rates of less than 30%. In the past, researchers and doctors believed that children with high-grade glioma were unfortunate individuals affected by a malignant disease that occurs more often in adults. Therefore, the understanding acquired from adults was applied to children suffering from the disease. However, although treatments for high-grade glioma have achieved modest improvements in survival rates in adults, in children these treatments did not make any difference [103,104]. In 2012, two groups—Schwartzentruber et al. and Lindroth et al.—showed that mutations in genes that encode histones, the spool-like proteins around which DNA is wound, contribute to high-grade glioma [105,106]. Understanding of diffuse midline gliomas, including diffuse intrinsic pontine glioma, which constitutes 75–80% of pediatric brain stem tumors, has been revolutionized by the discovery of novel mutations such as H3K27. This led to WHO reclassification of pediatric high-grade glioma in 2016 [103,107,108].

The third most common pediatric brain tumor is ependymoma. Anaplastic ependymoma and the posterior fossa variant remain the most formidable subtypes of pediatric ependymoma, both associated with a poor prognosis [109]. These subtypes have been suggested to originate from regional radial glial-like cells. Transcriptomics and DNA methylation profiling have both suggested that subtypes of ependymoma reflect the cell of origin. Ependymoma is a chemotherapy-resistant brain tumor, and despite genomic sequencing, effective molecular targets have not been identified [110].

Currently, the main therapeutic strategy for pediatric brain tumors is surgery, when it is possible. This is followed by chemotherapy and, rarely, radiotherapy, which can have severe side effects. Understanding of the molecular mechanisms of pediatric brain tumors remains limited, but molecular characterization could represent a new fundamental tool in the development of therapies for these tumors [111].

Despite advances in neuroimaging, surgical technology, conformal radiotherapy delivery, and conventional chemotherapy, current treatments for pediatric brain tumors are not prolonging patient survival. Therefore, there is an urgent need for more effective therapeutic approaches.

### 4.2. Potential Aptamer Targets in Pediatric Brain Tumors

Despite the remarkable effort of researchers to find effective treatments for pediatric brain tumors, the prognosis continues to be grim, even with targeted therapy. Nearly 40% of pediatric patients develop treatment resistance and relapse, and the morbidities and long-term sequelae of systemic therapy remain concerning. Research endeavors continue to identify oncogenic drivers of pediatric brain tumors, with some significant advances [99]. Molecular profiling has highlighted biological differences between pediatric and adult brain tumors. In particular, pediatric high-grade gliomas are characterized by recurrent mutations in the histone gene *H3F3A*, *TP53*, and the histone chaperone protein ATRX [107,112], leading to epigenetic changes in genes involved in tumor development [113]. Mutation, amplification, and upregulation of PDGFRα, associated with poor survival, has been reported in 15–39% of pediatric patients with high-grade glioma [114].

In recent years, profiling of potential druggable targets has surged, with the hope of improving treatment and preventing relapse and disease progression of pediatric central nervous system tumors. However, this has not translated into clinical success [115,116]. Unfortunately, the development of clinical trials using locoregional infusions of biologic agents targeting key oncogenic drivers has remained suboptimal in pediatric neuro-oncology. With the fast-paced elucidation of novel molecular targets in pediatric brain tumors, methods to enhance regional drug delivery into the central nervous system are warranted, to circumvent the alarming toxicities associated with systemic therapy and improve drug delivery.

Increased knowledge of the molecular mechanisms behind the pathogenesis of pediatric brain tumors, as well as identification of novel aptamers and/or adaptation of aptamers currently used in adult GBM, might open a novel research field; so far this remains unexplored. Already well-characterized aptamers for adult brain tumors may be useful in preclinical studies for the treatment of pediatric brain tumors. Genomic profiling reveals a wide landscape of genetic alterations, aberrant DNA methylation, histone modification patterns, and disorganized chromatin architecture involved in the pathogenesis of pediatric brain tumors, as well as in adult brain cancers.

As noted above, alterations of EGFR expression or of the ErbB signaling pathway are involved in tumor growth, angiogenesis, invasion, metastasis, and apoptosis inhibition. Truncation, overexpression, and amplification of EGFR have been reported as drivers in various types of pediatric brain tumors [117,118].

Another actionable gene is PDGFR, a cell surface tyrosine kinase receptor composed of PDGFRα and β. It regulates tumor progression, cell proliferation, invasion, angiogenesis, and neuronal differentiation. Alterations of the receptor, as well as its ligands, have been observed in gliomas, medulloblastoma, and ependymoma [117,118].

AXL, a receptor tyrosine kinase belonging with its homologs Tyro3 and Mer in the TAM (Tyro3-AXL-Mer) receptor kinase subfamily, is involved in epithelial-to-mesenchymal transition in adult GBM and has now been identified as a novel therapeutic target in diffuse intrinsic pontine glioma, correlating with HIST1H3B mutation. AXL overexpression has been shown to be involved in the regulation of GBM tumor growth through mechanisms that promote survival and growth of glioma stem cells and immunosuppressive signals [119,120]. This unique receptor is now being evaluated in pediatric brain tumors.

Another potential target is vascular endothelial growth factor (VEGF), which regulates angiogenesis and vascular permeability, and during hypoxic states, VEGF restores oxygen tension through hypoxia-response elements. Nowadays, targeted therapy against VEGF represents the most effective treatment to prevent tumor progression [117].

The Ras/mitogen-activated protein kinase pathway plays a key role in oncogenesis, proliferation, and survival, through activation of a downstream effector named Ras. Ras mutations and activation of Ras through overexpression or amplification of the growth factor receptor have been implicated in the development of pediatric low-grade glioma [117].

Integrins, particularly αvβ3 and αvβ5, are involved in tumor invasion, proliferation, and angiogenesis. Integrins mediate cell-matric adhesion and cell migration in the proper location between extracellular and intracellular compartments. Integrins can interact with various tyrosine kinase receptors, implementing their signaling pathways and phosphorylating them in the absence of their own ligands [117].

Recent studies have identified the Hedgehog (HH) pathway as an oncogenic driver for pediatric brain cancers, given this pathway’s important role in embryonic development, stem cell maintenance, cell differentiation, tissue polarity, and cell proliferation. Through activation of the Patched1 (PTCH1) receptor, Hh signaling controls transcription of target genes involved in cell survival, proliferation, and differentiation. In vertebrates, the activation of Hh signaling involves three ligands that bind the PTCH1 receptor and release its inhibitor, Smoothened (SMO). In turn, the activated SMO binds to SUFU and induces nuclear translocation of Hh pathway transcription regulators, modulating the expression of downstream targets. More than 20% to 30% of pediatric brain tumors have been reported to display activation of the HH pathway. In particular, germline mutations in Patched1 have been found to be responsible for somatic mutations in medulloblastoma, and 50% of medulloblastomas present loss of PTCH1, loss of SUFU, or gain-of-function of SMO [121,122].

The Notch signaling pathway is an important player in embryogenesis and carcinogenesis, as well as stem cell maintenance, in pediatric brain tumors. Notch1 has been observed to be important in the development of cancerous cells and stem cell maintenance, Notch2 was found to be upregulated in the hypothalamochiasmatic region, and Notch3 was shown to induce gliomas in the eye and optic nerve in murine models of pediatric low-grade astrocytoma [123].

SRY (sex determining region Y)-box 2, also known as SOX2, is upregulated in pediatric high-grade glioma and amplified in pediatric cell lines. Moreover, high levels of SOX2 have been detected in diffuse intrinsic pontine glioma tissues, consistent with the role of this protein in the maintenance of tumor stem cells. SOX2 is also expressed in Sonic Hedgehog-associated medulloblastoma, preferentially in adolescent and adult cases [124].

Another oncogenic driver is the isocitrate dehydrogenase enzymes, which have been reported to be involved in the progression of several cancers, including pediatric brain tumors. Isocitrate dehydrogenase 1 (IDH1) and IDH2 enzymes catalyze the conversion of isocitrate to α-ketoglutarate in the citric acid cycle; however, when mutations occur, these enzymes produce 2-hydroxyglutarate, an oncometabolite, which then accumulates and promotes cancer development through a wide spectrum of genomic alterations and DNA methylation changes. The most common somatic mutations of IDH1 result in replacement of arginine at position 132 by histidine (p.R132H), observed in diffuse grade II–III gliomas and secondary GBM, but rarely in primary GBM. Mutations in the *IDH2* gene (at 15q26.1), encoding the mitochondrial isocitrate dehydrogenase 2 (NADP+) enzyme, have also been detected in gliomas, but at a lower frequency [125,126].

Histone proteins are among the most conserved proteins in eukaryotes. They are involved in DNA packaging, allowing the DNA to fit inside the nucleus of a single cell. There have been many reports of the involvement of histone mutations in human disease. In 2012, a somatic heterozygous mutation in the gene encoding histone variant H3.3 and histone H3.1 was identified in pediatric GBM and diffuse intrinsic pontine glioma [127].

Another actionable target is the Ephrin receptor family. Deregulation of Eph/ephrin expression has been associated with an invasive phenotype, increased cell invasion, and the formation of metastases in medulloblastoma and ependymoma [128,129]. Eph/ephrin signaling is implicated in the development of several types of cancers and is involved in cell adhesion to substrates, as well as angiogenesis, migration, invasion, and growth. EphB2 and its ligand ephrin-B1 have been shown to be highly expressed in medulloblastoma tissue samples, and EphA2, EphB2, and EphB4 are overexpressed in medulloblastoma cell lines. EphB2 has been shown to play a pivotal role in the activation of pathways involved in cell adhesion and invasion in medulloblastoma [128].

IL13-PE38QQR, a recombinant cytotoxic chimera of human interleukin 13 and the enzymatically active portion of pseudomonas exotoxin A, has been shown to target glioma cell lines in preclinical studies. These findings have now been tested in a completed phase I trial using convection-enhanced delivery to evaluate the safety and efficacy of this treatment in patients with diffuse intrinsic pontine glioma [104].

Although aptamer-based technologies have been used for the treatment and imaging of adult brain tumors, these technologies have not made their way into pediatric neuro-oncology. With the continued poor survival outcomes in many pediatric brain tumors, novel therapies using aptamers (both from a therapeutic and imaging perspective to better understand the trafficking of infused biologic agents) need to be studied further. Preclinical data have already demonstrated the clinical efficacy and safety of aptamers in various tumors, and the evolution of this technology in adult neuro-oncology is now well understood. These findings need to be translated into the clinic in pediatrics (Figure 4) to improve the prognosis and survival of children harboring these formidable tumors. Studies over the next few years should evaluate steps to identify safe and efficacious aptamers that can target novel oncogenic biomarkers that induce the formation of these fatal neoplasms.

Preclinical scientific endeavors that can lead to a phase I study are warranted. These endeavors can evaluate the pharmacokinetic and pharmacodynamic properties, absorption, distribution, metabolism, and excretion of the test potential therapeutic agent. Once the test potential therapeutic agent has been shown to be active in animal models, safety evaluation is required prior to approval to proceed to a phase I clinical trial. The goals of phase I clinical trials are to establish the safety, pharmacokinetics, and potentially the biologically active dose of the new drug or investigational medicinal product. In most cases, two animal species are required for safety studies, one rodent and one non-rodent animal species such as monkeys; dosing is adjusted to body surface area. The ultimate goal is to determine the safety and quality of the investigational medicinal product. Single- and/or multiple-dose studies in mice will potentially shed light on the maximal tolerated dose and the dose-limiting toxicity and may uncover unexpected side effects. These studies help to identify target organs and the margin of safety in terms of the dose not observed to be associated with adverse effects [122]. Once the dose-limiting toxicity and maximal tolerated dose are identified in mice, a second species such as monkeys may be used for a “relevant” safety study, whereby the schedule proposed for the phase I human clinical trial is studied. This schedule may be based on results of the preclinical activity studies. (Adapted from FDA Exploratory IND Studies Guidance for Industry, Investigators, and Reviewers January 2006.) This translational scientific endeavor could help change the treatment paradigm and define a novel therapy using aptamers for pediatric high-grade gliomas—an unmet need in pediatric neuro-oncology

## 5. Conclusions

Increasing evidence indicates that aptamers are promising anticancer drugs owing to their unique characteristics (high-affinity binding, low immunogenicity, and ease of synthesis and chemical modification). The increased number of peer-reviewed publications, U.S. patents, and companies investing in aptamer development are projected to increase the worldwide aptamer market. Nowadays, these multifunctional agents are widely studied among scientists. Various molecules involved in tumor progression and metastasis have been designed to target a wide spectrum of oncogenic biomarkers. Several aptamers are in clinical trials for therapeutic and diagnostic application, and one aptamer has already been approved by the U.S. Food and Drug Administration. AS1411, a PEGylated DNA aptamer G-rich quadruplex targeting nucleolin, and NOX-A12, a PEGylated Spiegelmer RNA aptamer targeting chemokine CXCL12, as well as many others, have reached clinical trials as diagnostic or anticancer agents (Table 3). In preclinical studies, aptamers have shown effectiveness and good safety profiles. Chemical modifications have been shown to be the main challenge to translating aptamers to clinical applications [130,131]. These biological enrichments and successful clinical translation in adult neuro-oncology reaffirms the need to test aptamers in clinical studies of pediatric brain tumors.

## Figures and Tables

**Figure 1 cancers-12-02889-f001:**
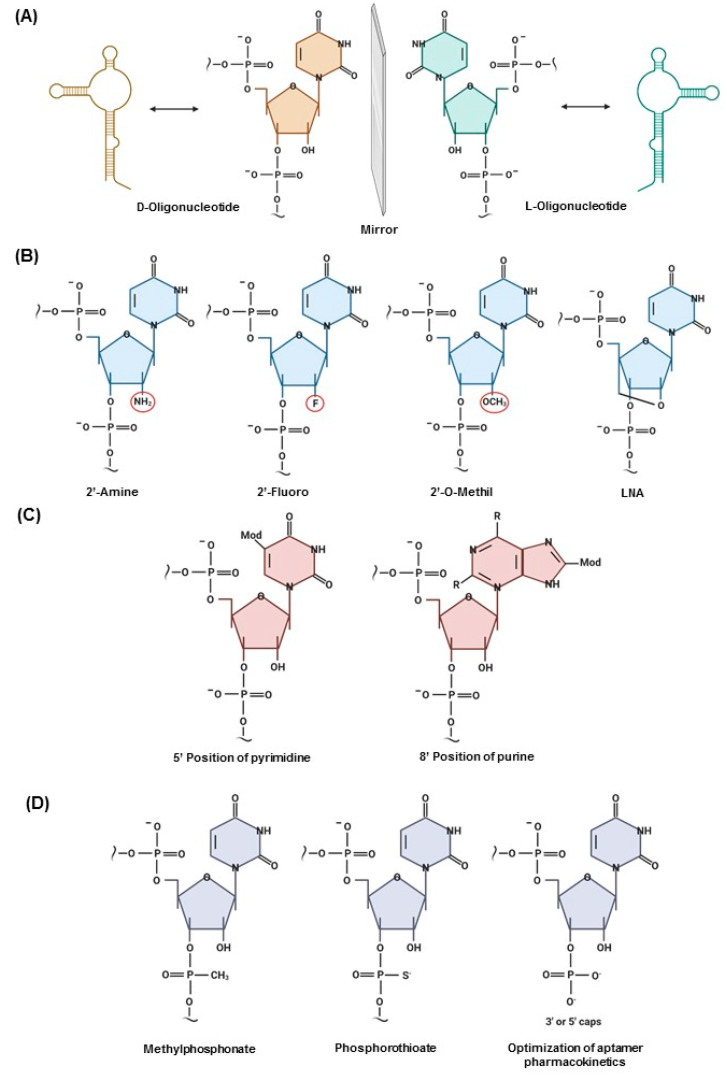
Post-Systemic Evolution of Ligands by Exponential enrichment (SELEX) modifications. (**A**) Spiegelmers aptamer strategy: chiral inversion transition resulting in mirror image. (**B**) Modifications of the sugar ring: replacement of 2′ positions with 2′-NH2, 2′-F, and 2′-O-CH3, as well as locked nucleic acids (LNA). (**C**) Modifications of bases in the 5′ position of pyrimidine and 8′ position of purine. (**D**) Modifications of the phosphodiester linkage and pharmacokinetic optimization by capping at the 5′- or 3′-end.

**Figure 2 cancers-12-02889-f002:**
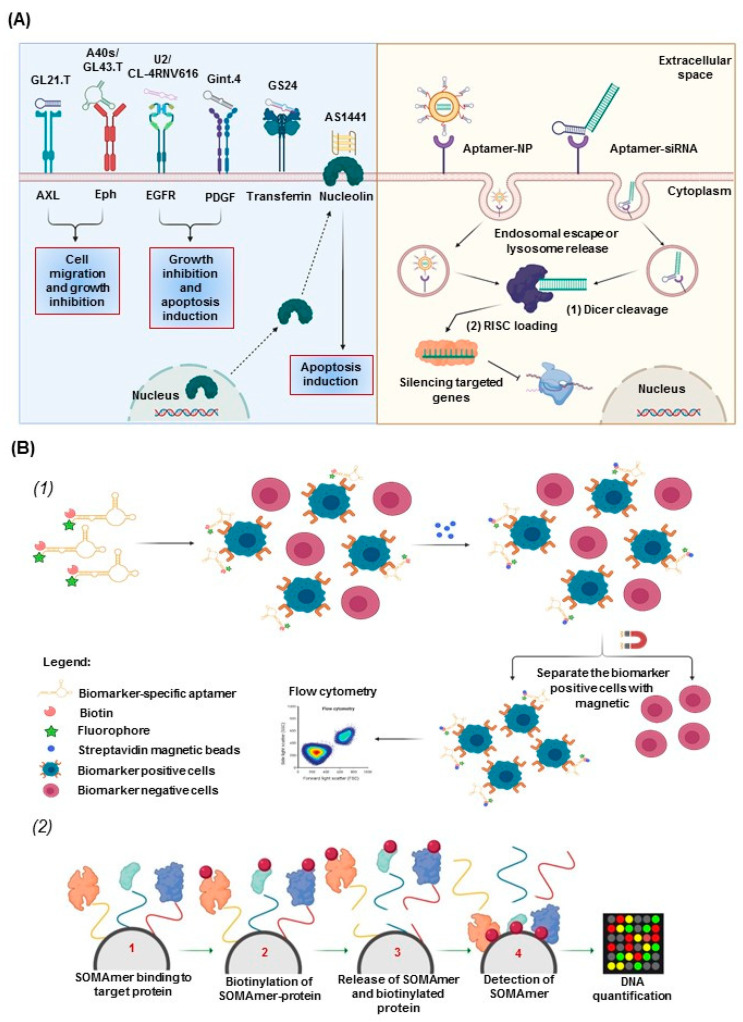
Aptamers as therapeutics and diagnostic tools. (**A**) Therapeutic aptamers alone (left panel) and aptamer-targeted drug delivery through receptor-mediated endocytosis (right panel): (1) dicer cuts the long double-strand RNA to form small interfering RNA (siRNA); (2) siRNA is loaded into of the RISC complex for further recognition, cleavage, and mRNA degradation. (**B**) Aptamers as diagnostic tools, as shown by (1) a schema of the OLIGOBIND strategy and (2) a schema of the SOMAscan (based on slow off-rate modified aptamer (SOMAmer) procedure for DNA quantification).

**Figure 3 cancers-12-02889-f003:**
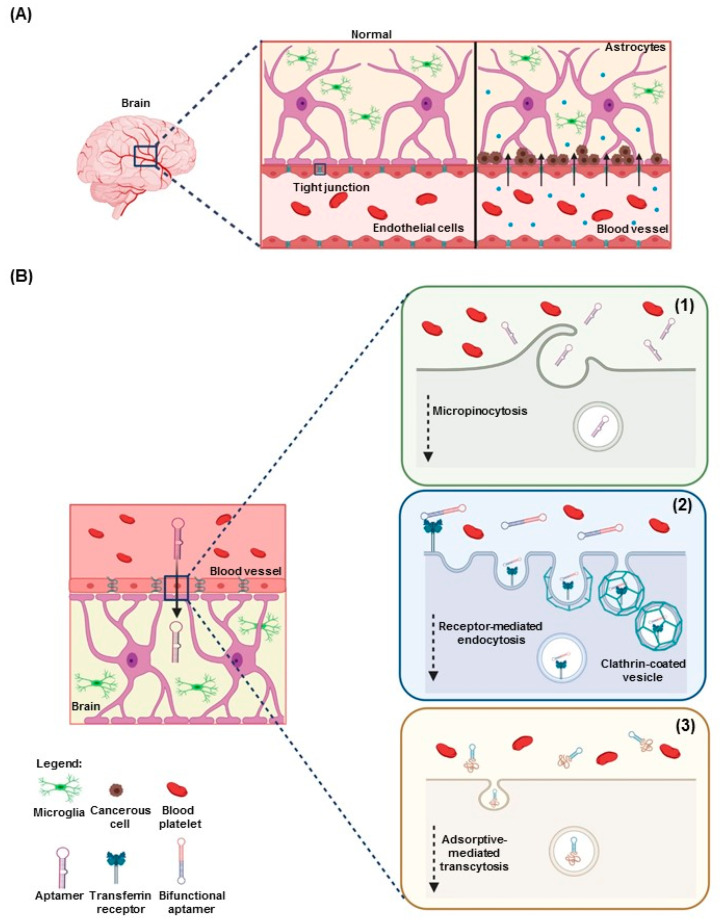
Mechanisms used by aptamers to penetrate the blood–brain barrier (BBB). (**A**) Left panel shows the normal BBB, in which the tight junctions are closed, blocking penetration through the BBB. Right panel shows tumor-induced BBB disruption, allowing penetration through opened tight junctions. (**B**) Proposed mechanism used by aptamers to cross the BBB: (1) micropinocytosis, (2): receptor-mediated endocytosis, and (3) adsorptive-mediated transcytosis through cell-penetrating peptide shuttles.

**Figure 4 cancers-12-02889-f004:**
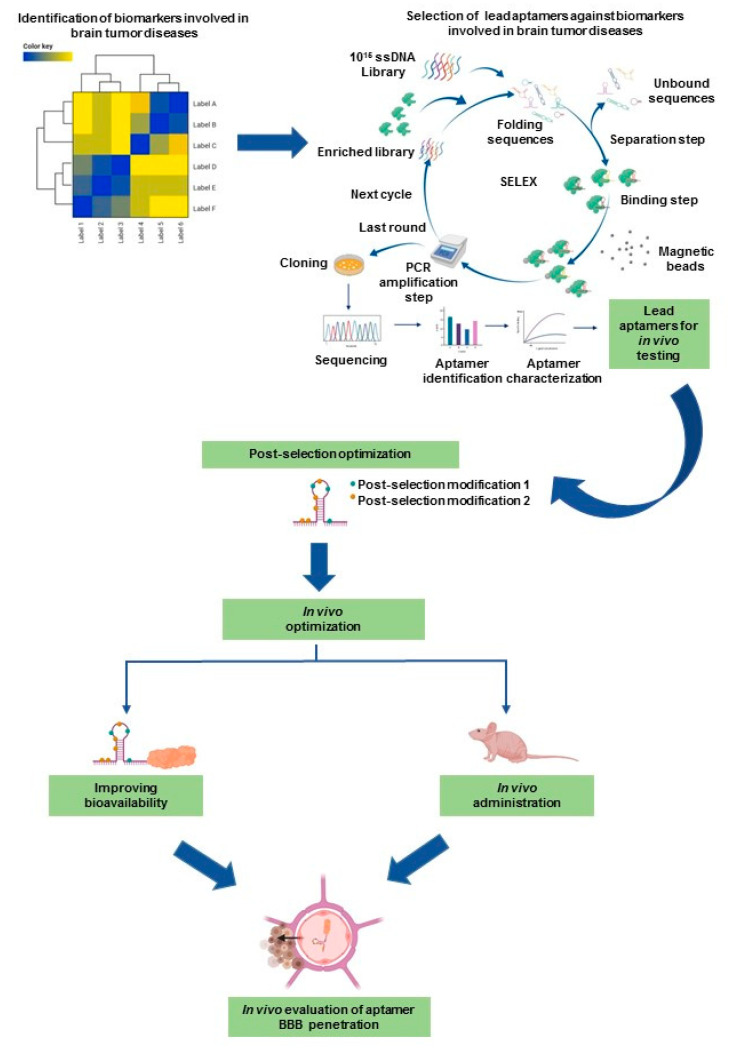
Schema for potential development of aptamers in neuro-oncology and pediatric brain tumors.

**Table 1 cancers-12-02889-t001:** Aptamers as therapeutic tools in neuro-oncology reported in the literature in the past 5 years.

Aptamer Name	Year	Molecular Target	Aptamer Role	SELEX Method	K_D_ (nmol/L)	Composition	Therapeutic Application	References
Gint4.T	2020	PDGFRβ	Antagonist	Cell-based SELEX	9.6	2′-Fluoro RNA; conjugated with STAT3 siRNA	Reduce GBM tumor growth and relapse	[69]
GL21.T	2020	AXL	Antagonist	Cell-based SELEX	13	2′-Fluoro RNA; conjugated with miR-10b	Reduce GBM tumor growth and relapse	[69]
4-1BB–OPN	2019	CD8^+^/OPN	Agonist/antagonist	SELEX primer/in vitro SELEX	40/18	2′-Fluoro RNA/RNA bispecific aptamer	CD8+ T cell activation/block M0 and M2 macrophage migration	[70]
Gint4.T/GMT8	2019	PDGFRβ/U87 cell line	Antagonist/unknown	Cell-based SELEX/whole-cell SELEX	9.6/unknown	2′-Fluoro RNA/DNA nanocarrier	Cross BBB and deliver paclitaxel	[71]
PDR3	2019	PDGFRα	Antagonist	Protein-based SELEX	0.25	RNA	Reduce GBM tumor growth	[72]
CL-4RNV616	2019	EGFR	Antagonist	Cell-based SELEX	18.24	2′-O-methyl RNA and DNA	Reduce cell proliferation	[73]
AS1411/GS24	2019	Nucleolin/transferrin receptor				G-rich DNA/DNA nanocarrier	Cross BBB and deliver TMZ, reduce drug resistance	[74]
A15	2018	CD133	In vivo SELEX			RNA/dual-targeting ligand/nanoparticles	Cross BBB/deliver siRNA	[75]
TfR Aptamer	2018	Transferrin Receptor			365	Conjugated with RNV541 DNAzyme/antimiRzymes chimera	Suppress miR-21 expression in U87MG malignant glioblastoma	[76]
Gint4.T	2018	PDGFRβ	Antagonist	Cell-based SELEX	9.6	2′-Fluoro RNA; conjugated with STAT3 siRNA	Inhibit cancer cell survival and migration.	[50]
Aptamer-like peptide	2018	EDB-fibronectin			16	APTEDB-PEG2000-DSPE	Liposome-based nanoparticle platform for systemic siRNA delivery	[77]
AS1411	2017	Nucleolin				Poly (L-c-glutamylglutamine)–paclitaxel nanoconjugates	Increased median survival time of GBM tumor-bearing mice	[78]
Gint4.T	2017	PDGFRβ	Antagonist	Cell-based SELEX	9.6	Conjugated with BODIPY@PNPs	Cross BBB and deliver drugs	[79]
GL43.T	2016	EphB3/2	Antagonist	Cell-based SELEX	433.5	2′-Fluoro RNA	Inhibit cell migration	[80]
GL21.T/Gint4.T	2016	AXL/PDGFRβ	Antagonist	Cell-based SELEX	13/9.6	Conjugated with miR-137 and antimiR-10b	Target glioma stem-like cells	[81]

**Table 2 cancers-12-02889-t002:** Aptamers for diagnostics and imaging in neuro-oncology as reported in the literature in the past 5 years.

Aptamer Name	Year	Molecular Target	K_D_ (nmol/L)	SELEX Method	Composition	Diagnostic or Imaging Application	References
TD05	2020	CD20+ B cells	256	Cell-based SELEX	Conjugated with Alexa-488	Diagnostic (intraoperative tumor-specific) Imaging (ex vivo fluorescence microscopy)	[82]
A40s	2019	EphA2	41.92	Cell-based SELEX	2’-Fluoro RNA conjugated with siRNA or miRNA	Imaging (confocal fluorescence microscopy)	[83]
H02	2019	Integrinα5β1	72–277.8	Protein SELEX	2’-Fluoro RNA conjugated with Cy5, Alexa-564	Imaging (confocal fluorescence microscopy)	[84]
WYZ-41a WYZ-50a	2018	A172 cells	75.27–168.56	Cell-based SELEX	DNA conjugated with Cy5, FITC	Imaging (confocal fluorescence microscopy)	[85]
A15	2018	CD133		In vivo SELEX	RNA/dual-targeting ligand/nanoparticles	Imaging (in vivo bioluminescency)	[75]
Anti-EGFR	2018	EGFRvIII			2′-Fluoro RNA	Dynamic morphology	[86]
QD-A32 Apt,	2017	EGFRvIII			DNA conjugated with streptavidin-PEG-CdSe/ZnS QDs	Imaging (in vivo imaging)	[87]
Quenched-TD05	2015	CD20+ B cells	256	Cell-based SELEX	FRET-based switchable aptamer	Imaging (confocal fluorescence microscopy) Diagnostic (intraoperative diagnoses)	[88]
GBI-10	2015	Tenascin-C	150	In vitro SELEX	Gadolinium-loaded liposomes	Diagnostic (magnetic resonance imaging)	[89]
Anti-EGFR	2015	EGFRvIII			2′-Fluoro RNA	Dynamic morphology	[90]

**Table 3 cancers-12-02889-t003:** Aptamers as a “*magic bullet*” for therapeutic/diagnostic strategies in clinical trials.

Aptamer Name	Molecular Target	Composition	Administration Route	Therapeutic Applications	Clinical Trials	Status	Clinical Trials.gov Identifier
AS1411 (AGRO001)	Nucleolin	26-mer G-rich DNA	Intravenous	Glioblastomas, acute myeloid leukemia, metastatic renal cell carcinoma	Phase II, phase 1, and phase II	Completed, completed, and unknown	NTC01034410, NTC00881244, and NTC00740441
Anti-VEGF PEGylated aptamer (EYE001)	VEGF		Intravitreal injection	Retinal tumors in patients with Von Hippel-Lindau syndrome	Phase I	Completed	NCT00056199
NOX-A12	Angiogenic chemokine (C-X-C motif) ligand 12 (CXCL12, also known as SDF-1α)	45-mer L-RNA with 3′-PEG (Spiegelmer)	Intravenous	Multiple myeloma, non-Hodgkin lymphoma, leukemia, metastatic colorectal cancer	Phase II, phase II, and phase I	Completed, completed, and unknown	NTC01521533, NTC01486797, and NTC03168139
Aptamer sensors	Discover urinary biomarker(s)	Electro-phage and colorimetric aptamer sensors for clinical staging and monitoring by FRET system	Urine test	Bladder cancer		Recruiting patients	NCT02957370
68Ga-Sgc8	PTK7	41-mer DNA bi-functional aptamer; diagnostic performance and evaluation efficacy of a novel PTK7 positron emission tomography radiotracer 68Ga-SGC8	Intravenous	Colorectal cancer	Early-phase I		NTC03385148
Aptamers to target tumor cells In the laboratory	Unknown	Tailored neoadjuvant epirubicin and cyclophosphamide and nanoparticle albumin bound paclitaxel for newly diagnosed breast cancer	Tumor tissue in vitro	Breast cancer	Phase II		NCT01830244
X-aptamers library	Unknown	Novel proteomic biomarkers for HCC patients treated with Lipiodol TACE using beads-based X-aptamer library, then validate and create a biomarker panel that can be used to predict the outcome of patients with hepatocellular carcinoma after treatment with Lipiodol TACE	Blood test	Hepatocellular carcinoma	July 2020	Not yet recruiting	NCT04459468

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
