# Peer review of "Aptamers: Novel Therapeutics and Potential Role in Neuro-Oncology"

_cancers, 2020, doi:10.3390/cancers12102889_

Round 1

Reviewer 1 Report

Paola A, Gabriel LP et.al have in their review discussed about the biology of aptamers and the challenges and potential therapeutic role of aptamers in pediatric Neuro-Oncology. I commend the authors for attempting to write this review. However, I find this review more focused on the biology of the aptamers and less on its use in Neuro-oncology in general and pediatric brain tumors in particular. This review reads like "cut and paste" from all the exhaustive references that have been cited by the authors. The incorporation of following suggestions would go a long way in making this review more concise and appropriate for publication in Cancers.

1) Title reads "Aptamers: novel therapeutics and potential role in Neuro-Oncology", and the authors show data in one single table (Table-2). I recommend that authors provide more pre-clinical and clinical data (see latest review from Silvia Nuzzo's team from Italy).

2)  There are a number of recent and excellent reviews available on aptamers for specific use a) as a tool in Neuroscience: Wolter and Mayer (J of Neuroscience). b) Aptamers as therapeutics (Sullenger et.al Ann Rev Pharmatox). c) aptamer modifications-Adachi/ Nakamura -Molecules) etc. I suggest that headings 2, 3, 4 and 5 be condensed into one and written more concisely. Table-1 can be modified or completely excluded as it does not provide anything novel. I suggest we can retain fig-1, with addition of aptamers as a therapeutic tool.

3) heading-6 should be expanded and include pre-clinical and clinical data from adult studies (Nuzzo et.al, Cancers, ). I understand that there is virtually no data in Pediatric neuro-oncology. Figure-2 can be retained. However, Table -2 needs extensive revision with seperate headings if possible (Diagnostic / Therapeutic / Imaging etc).

4) Figure-3 can be used in general for Neuro-oncology (not only for pediatric brain tumors). Heading -7 needs complete revision and Clinicians corner should give more directions in terms of possible adaptations of adult aptamer based clinical trials in the pediatric populations. I suggest limiting table-3 to be expanded and limited only to Cancers (both CNS and Non-CNS cancers).

I once again commend the authors for attempting to write a review on this emerging and exciting topic.     

Author Response

Gabriel Lopez-Berestein, MD.

Professor

Department Experimental Therapeutics

Phone: 713-792-3508 

Email:glopez@mdanderson.org

Address: 1515 Holcombe Blvd.,

Unit 1950, Houston, Texas 77030

Dear Reviewer:

We appreciate the opportunity to revise our manuscript, ID: cancers-901527, entitled “Aptamers: Novel Therapeutics and Potential Role in Neuro-Oncology”. We would like to thank the reviewers for the valuable comments that they provided to enrich our review article.  Below are the detailed response to all reviewers’ queries.

Reviewer Comments:

Reviewer #1

  • Comment 1:

“Paola A, Gabriel LP et.al have in their review discussed about the biology of aptamers and the challenges and potential therapeutic role of aptamers in pediatric Neuro-Oncology. I commend the authors for attempting to write this review. However, I find this review more focused on the biology of the aptamers and less on its use in Neuro-oncology in general and pediatric brain tumors in particular. This review reads like "cut and paste" from all the exhaustive references that have been cited by the authors. The incorporation of following suggestions would go a long way in making this review more concise and appropriate for publication in Cancers.”

  • Response 1:

We would like to thank the reviewer for its positive comment regarding our manuscript. The manuscript has been extensively revised. As the reviewer suggested, we incorporated heading 2-3-4-5 in one (Please refer to new heading 2) and we made it more concise. However, we believe that the aptamer biology is needed, as general overview, in order to explain why aptamer technology can be apply to brain tumor treatment and diagnosis. Moreover, we extend the aptamers in neuro-oncology (Please refer to new heading 3) and pediatric neuro-oncology paragraphs (Please refer to new heading 4).

  • Comment 2:

Title reads "Aptamers: novel therapeutics and potential role in Neuro-Oncology", and the authors show data in one single table (Table-2). I recommend that authors provide more pre-clinical and clinical data (see latest review from Silvia Nuzzo's team from Italy).”

  • Response 2:

We appreciate the comment and we have addressed it including more preclinical and clinical data. Table 2 (Please now refer to new Table 1) has been completely revised. We have reported the more recently studies published in the last 5 years, regarding aptamers as therapeutics (Please refers to new Table 1 A) and aptamers as diagnostic and imagining (Please refer to new Table 1 B). Moreover, the review article from Nuzzo et. al has been included in the references (Please refer to reference # 74).

  • Comment 3:

“There are a number of recent and excellent reviews available on aptamers for specific use a) as a tool in Neuroscience: Wolter and Mayer (J of Neuroscience). b) Aptamers as therapeutics (Sullenger et.al Ann Rev Pharmatox). c) aptamer modifications-Adachi/ Nakamura -Molecules) etc. I suggest that headings 2, 3, 4 and 5 be condensed into one and written more concisely. Table-1 can be modified or completely excluded as it does not provide anything novel. I suggest we can retain fig-1, with addition of aptamers as a therapeutic tool.”

  • Response 3:

We would like to thank the reviewer for its effort in reading our manuscript and suggesting more valuable references to enrich our manuscript. Aptamers as Valuable Molecular Tools in Neurosciences by Wolter and Mayer, J of Neuroscience (Please refer to reference # 17); Aptamers as therapeutics by Sullenger et.al Ann Rev Pharmatox (Please refer to reference # 51); aptamer modifications-Adachi/ Nakamura –Molecules (Please refer to reference #40) have been cited in our review article. Heading 2-3-4-5 have been condensed into one and Aptamers as therapeutics has been incorporated into figure 1 (Please refer to new Figure 2A, left panel). Previous Table 1 has been excluded.

  • Comment 4:

“Heading-6 should be expanded and include pre-clinical and clinical data from adult studies (Nuzzo et.al, Cancers,). I understand that there is virtually no data in Pediatric neuro-oncology. Figure-2 can be retained. However, Table-2 needs extensive revision with seperate headings if possible (Diagnostic / Therapeutic / Imaging etc).”

  • Response 4:

We have addressed the comment by expanding the section 6 (Please refer to new heading 3). As the reviewer suggested, we included more pre-clinical and clinical data from adult glioblastoma. We cited the review article “The Role of RNA and DNA Aptamers in Glioblastoma Diagnosis and Therapy: A Systematic Review of the Literature. By Nuzzo et al (Cancers)” (Please refer to reference # 74). Table-2 (Please refer to new Table 1 A and B) has been revised and we separated aptamers as therapeutics (Please refer to new Table 1 A) and aptamers as diagnostic and imaging (Please refer to new Table 1 B).

  • Comment 5:

“Figure-3 can be used in general for Neuro-oncology (not only for pediatric brain tumors). Heading -7 needs complete revision and Clinicians corner should give more directions in terms of possible adaptations of adult aptamer based clinical trials in the pediatric populations. I suggest limiting table-3 to be expanded and limited only to Cancers (both CNS and Non-CNS cancers).”

  • Response 5:

We value your accurate comment and we have included “neuro-oncology” in the Figure legend and deleting “pediatric” into the text of previous Figure 3 (Please refer to new Figure and Figure legend 4). Heading 7 has been carefully revised (Please refer to new heading 4). We have included a new Table where we depicted the latest clinical trials in pediatric brain tumors (Please refer to new Table 3).

Comment 6:

“I once again commend the authors for attempting to write a review on this emerging and exciting topic.”

  • Response 6:

We would like to thank the reviewer for the positive comment regarding our manuscript. We deeply believe that aptamers can make the difference as Novel Therapeutics and hold a great Potential in Neuro-Oncology.

Please let me know if I can supply any additional information regarding our manuscript.

With my best regards,

Dr. Gabriel Lopez-Berestein,

Reviewer 2 Report

RE: Aptamers: Novel Therapeutics and Potential Role in 2 Neuro-Oncology

Overall the review reads a bit more like a book chapter, than a scientific review. The content is good but overly wordy. The figures are nice but overly simplistic. The aspects on pediatric brain tumors should be improved.

1) In sections 6 and 7, on neuro-oncology, most of the writing is very general neuro-onc topics (nomenclature, BBB etc.), then when the authors get the the aptamer part, it is very brief such as "Cheng et al investigated the potential of aptamers to penetrate the BBB in animal models, 264 demonstrating that aptamers can internalize into endothelial cells through target receptor-mediated 265 transport mechanisms [31]." Overall more discussion is needed on mansucripts using aptamers in neuro-oncology.

2) Article should be revised throughout for wordiness and meaning.
Some examples:
-In the abstract "Although a few aptamer-related translational studies have been performed in adult glioblastoma, the use of aptamers in pediatric 21 neuro-oncology remains largely unexplored." - "Few" and "largely" unexplored have roughly the same meanings

-"Among pediatric solid cancers, brain tumors are the leading cause of death in children." - there is no need to reference children twice in this one sentance. Rather, "brain tumors are the leading cause of cancer-related death in children"

-Line 29- "Theoretically, nucleic acids may have a multifunctional nature." - "theroetically" and "may" are saying the same thing

-line 236- Glial tumors, which arise from 235 glial cells, are the most prevalent type of adult brain tumor. There are various important subtypes of 236 gliomas, including diffuse gliomas, non-diffuse gliomas, and ependymomas, classified according to 237 the genotype of the origin glial precursor cell. - please revise for accuracy and wordiness.

-line 253- The clinical benefit of BBB disruption has not been established for the treatment of less sensitive tumors such as gliomas, but BBB disruption seems to increase survival in patients. - the words "seems" is akward. was it associated with survival in a small cohort or what? The entire section on BBB needs more citations.

-line 302- "Medulloblastoma has been the most extensively studied pediatric brain tumor from a genomic and epigenomic perspective" - better to say medulo is a "well studied..." or something like that.

3) Figures should be considered for more "complex" concepts like the 3 strategies for post-selex modification.

Author Response

Gabriel Lopez-Berestein, MD.

Professor

Department Experimental Therapeutics

Phone: 713-792-3508 

Email:glopez@mdanderson.org

Address: 1515 Holcombe Blvd.,

Unit 1950, Houston, Texas 77030

Dear Reviewer:

We appreciate the opportunity to revise our manuscript, ID: cancers-901527, entitled “Aptamers: Novel Therapeutics and Potential Role in Neuro-Oncology”. We would like to thank the reviewers for the valuable comments that they provided to enrich our review article.  Below are the detailed response to all reviewers’ queries.

Reviewer Comments:

Reviewer #2

  • Comment 1:

“Overall the review reads a bit more like a book chapter, than a scientific review. The content is good but overly wordy. The figures are nice but overly simplistic. The aspects on pediatric brain tumors should be improved.”

  • Response 1:

We would like to thank the reviewer for its comments and effort to improve our manuscript. We have made the review article more concise summarizing the aptamer biology part (Please refer to new heading 2). As the reviewer suggested, we added another figure regarding the post SELEX modifications (Please refer to new Figure 1) and we improve the previous Figure 1 (Please refer to new Figure 2A, left panel) including the section about aptamers as therapeutics. The pediatric brain tumor section has been carefully revised and expanded (Please refer to new heading 4). Moreover, we included a Table regarding the latest clinical trials reported for pediatric brain tumor (Please refer to new Table 3).

  • Comment 2:

“In sections 6 and 7, on neuro-oncology, most of the writing is very general neuro-onc topics (nomenclature, BBB etc.), then when the authors get the the aptamer part, it is very brief such as "Cheng et al investigated the potential of aptamers to penetrate the BBB in animal models, 264 demonstrating that aptamers can internalize into endothelial cells through target receptor-mediated 265 transport mechanisms [31]." Overall more discussion is needed on manuscripts using aptamers in neuro-oncology.”

  • Response 2:

We appreciate your valuable comments and we have extensively revised the previous section 6 and 7 (Please refer to new heading 3 and 4). In new section 3, we have included two additional sections, the first one about aptamers as therapeutics (Please refer new heading 3.1 from line 271 -341) and the second one about diagnosis (Please refer new heading 3.2 from line 345- 380) in GBM. Previous Table 2 has been extensively revised (Please refer to new Table 1A and B). We generated de novo two tables, one for aptamer as therapeutics (Please refer to new Table 1A) reviewing the studies related to aptamers in the therapeutic field in the last 5 years; and the second table (Please refer to new Table 1B) reviewing the studies related to diagnostic and imagining field in the last 5 years, as well.

  • Comment#3:

“Article should be revised throughout for wordiness and meaning.

Some examples:

-In the abstract "Although a few aptamer-related translational studies have been performed in adult glioblastoma, the use of aptamers in pediatric 21 neuro-oncology remains largely unexplored." - "Few" and "largely" unexplored have roughly the same meanings.

-Among pediatric solid cancers, brain tumors are the leading cause of death in children." - there is no need to reference children twice in this one sentance. Rather, "brain tumors are the leading cause of cancer-related death in children.

- Line 29- "Theoretically, nucleic acids may have a multifunctional nature." - "theroetically" and "may" are saying the same thing.

- Line 236- Glial tumors, which arise from 235 glial cells, are the most prevalent type of adult brain tumor. There are various important subtypes of 236 gliomas, including diffuse gliomas, non-diffuse gliomas, and ependymomas, classified according to 237 the genotype of the origin glial precursor cell. - please revise for accuracy and wordiness.

- line 253- The clinical benefit of BBB disruption has not been established for the treatment of less sensitive tumors such as gliomas, but BBB disruption seems to increase survival in patients. - the words "seems" is akward. was it associated with survival in a small cohort or what? The entire section on BBB needs more citations.

- Line 302- "Medulloblastoma has been the most extensively studied pediatric brain tumor from a genomic and epigenomic perspective" - better to say medulo is a "well studied..." or something like that.”

  • Response #3:

We appreciate the comments provided by the reviewer and the great opportunity to address them in our review article

- Line 22: We addressed this point removing “largely”.

- Line 20: We addressed this point removing “in children”.

- Line 29: We addressed this point removing “may”.

- Line 236: We carefully revised this sentence. Please refer to line 234.

- Line 253: We carefully revised this sentence. Please refer to line 252. As the reviewer suggested, we included more references. Please refer to new references 67-68.

- Line 312: We carefully revised this sentence, we addressed the comment of the reviewer following its recommendations. Please refer to new line 389.

  • Comment#4:

“Figures should be considered for more "complex" concepts like the 3 strategies for post-selex modification.”

  • Response #4:

We have added a new figure (Please refer to new Figure 1) regarding the three strategies for aptamer post-SELEX modifications.

Please let me know if I can supply any additional information regarding our manuscript.

With my best regards,

Dr. Gabriel Lopez-Berestein,

Round 2

Reviewer 1 Report

Cristian and Gabriel et.al. have done a reasonably good job of the revision. However, there is still a lot to be revised and needs re-writing. My abridged comments are as follows:

1) All newly added text needs extensive English editing and revisions. There are too many grammatical /sentence construction errors for me to point it out.

2) The authors have done a good job of revising the biology of aptamers (especially Figure-1). However, Figure -2 needs revision both at the figure level and the legend. Except for the SOMAmer scan, which is used as a diagnostic, the rest of the figure is mostly for therapeutics ( either as a direct target (A) or as a delivery system (B,C). The figure has been mis-labelled and I do not see any D in the figure.   

3)   The meat of the review starts from Section-3 and Fig-3. Section 3.2 needs extensive revisions-along with table-1B. For eg: It is Imaging not (Imagining).

4) New text needs English revision in section 4.1. Table-3 is redundant and can be omitted.  Legend for Fig-4 needs revision: For eg: Schema for development of Aptamers as potential therapeutics in Neuro-Oncology.

5) I suggest combining Section 4.2 with Clinician's corner and explain in detail the potential targets in pediatric brain tumors.

6) Table-4 can be retained and updated to make it current. 

Author Response

Dear Reviewers:

We appreciate the opportunity to revise our manuscript, ID: cancers-901527, entitled “Aptamers: Novel Therapeutics and Potential Role in Neuro-Oncology.” We would like to thank the reviewers for the valuable comments that they provided to enrich our review article.  Below are detailed responses to all reviewer queries.

Reviewer Comments:

Reviewer #1

  • Comment 1:

“Cristian and Gabriel et.al. have done a reasonably good job of the revision. However, there is still a lot to be revised and needs re-writing. My abridged comments are as follows.”

  • Response 1:

We would like to thank the reviewer for the positive comment regarding our manuscript. The manuscript has been extensively revised, and we addressed the comments of the reviewer.

  • Comment 2:

All newly added text needs extensive English editing and revisions. There are too many grammatical /sentence construction errors for me to point it out.”

  • Response 2:

English has been extensively revised and edited by Editing Services in the Research Medical Library at MD Anderson Cancer Center.

  • Comment 3:

“The authors have done a good job of revising the biology of aptamers (especially Figure-1). However, Figure -2 needs revision both at the figure level and the legend. Except for the SOMAmer scan, which is used as a diagnostic, the rest of the figure is mostly for therapeutics ( either as a direct target (A) or as a delivery system (B,C). The figure has been mis-labelled and I do not see any D in the figure.”

  • Response 3:

We would like to thank the reviewer for their suggestions to improve the readability of our manuscript. Figure 2 has been revised and now includes two sections: (A) aptamers as therapeutic tools and (B) aptamers as diagnostic tools.

  • Comment 4:

“The meat of the review starts from Section-3 and Fig-3. Section 3.2 needs extensive revisions-along with table-1B. For eg: It is Imaging not (Imagining).”

  • Response 4:

The English has been revised by Editing Services in the Research Medical Library at MD Anderson Cancer Center.

  • Comment 5:

“New text needs English revision in section 4.1. Table-3 is redundant and can be omitted.  Legend for Fig-4 needs revision: For eg: Schema for development of Aptamers as potential therapeutics in Neuro-Oncology.”

  • Response 5:

We have deleted the original Table 3 as the reviewer suggested. We have also revised the legend for Figure 4.

Comment 6:

“I suggest combining Section 4.2 with Clinician's corner and explain in detail the potential targets in pediatric brain tumors.”

  • Response 6:

We have combined the Clinician’s corner section with potential targets in pediatric brain tumors. We have therefore revised heading 4.2.

  • Comment 7:

“Table-4 can be retained and updated to make it current.”

  • Response 7:

We have updated the previous Table 4; please now refer to Table 3.

Reviewer 2 Report

The revised manuscript by Dr. Paola et al.  on aptamer technology for neuro-oncology is significantly improved including: new figure 1 and the detailed description of studies using aptamers in neuro-oncology. While the content is good, still the review could use editing for wordiness, style, and english. I think 20% could be cut while still keeping true to the review and retaining almost all the current content. This would make it more appealing and useful for the readers. 

Some other comments:

-Please add in around line 81 what Selex used for. Also define "SELEX" at first use.

-Line 185- Provide better description of the Spiegelmers strategy.

-Please add a little more information in the figure 1 legend.

-line 219: I would remove this entire section as it seems out of place: "The concept of personalized and tailored drugs was coined more than 100 years ago by the founder of chemotherapy, Paul Ehrlich. His paradigm inspired advanced research in molecular biology and drug discovery, guiding the development of new therapeutic agents that are successfully applied in clinical trials [49]. Based on Ehrlich’s concept, therapeutic oligonucleotides represent an advanced class of “smart” drugs that can regulate gene expression patterns involved in the progression of many diseases [50]."

Line 255: Unclear how SOMAscan relates to aptamers.

Line 473- For the following section (pasted below), I would recommend including the words "when possible" when talking about surgery. Also radiotherapy is used. Often wait and see approach is also used. Finally, some current treatments are prolonging pediatric brain tumor survival.

"Currently, pediatric brain tumor treatments are represented by surgery, the main therapeutic strategy, following by chemotherapy and rarely radiotherapy, due the severe side effects. Molecular mechanisms, behind pediatric brain tumors remains still limited, but it could represent a new fundamental tool in the therapeutic perspective of these tumors [116].
The current treatments are not prolonging patient survival. There is an urgently need It is needed of for more effective therapeutic approaches. In Table 3, we reported the latest clinical trials for pediatric brain tumors."

Author Response

Dear Reviewers:

We appreciate the opportunity to revise our manuscript, ID: cancers-901527, entitled “Aptamers: Novel Therapeutics and Potential Role in Neuro-Oncology.” We would like to thank the reviewers for the valuable comments that they provided to enrich our review article.  Below are detailed responses to all reviewer queries.

Reviewer Comments:

Reviewer #2

  • Comment 1:

“The revised manuscript by Dr. Paola et al. on aptamer technology for neuro-oncology is significantly improved including: new figure 1 and the detailed description of studies using aptamers in neuro-oncology. While the content is good, still the review could use editing for wordiness, style, and english. I think 20% could be cut while still keeping true to the review and retaining almost all the current content. This would make it more appealing and useful for the readers.”

  • Response 1:

We would like to thank the reviewer for their comments. The English has been revised by Editing Services in the Research Medical Library at MD Anderson Cancer Center.

  • Comment 2:

“Please add in around line 81 what Selex used for. Also define "SELEX" at first use.”

  • Response 2:

We would like to thank the reviewer for this valuable comment. We have added an explanation of the use of SELEX, including its definition at first use (please refer to lines 103-108).

  • Comment#3:

“Line 185- Provide better description of the Spiegelmers strategy.”

  • Response #3:

We have revised our description of the Spiegelmers strategy. Please refer to lines 185-195.

  • Comment#4:

“Please add a little more information in the figure 1 legend”

  • Response #4:

We have revised the legend for Figure 1.

  • Comment#5:

“line 219: I would remove this entire section as it seems out of place: "The concept of personalized and tailored drugs was coined more than 100 years ago by the founder of chemotherapy, Paul Ehrlich. His paradigm inspired advanced research in molecular biology and drug discovery, guiding the development of new therapeutic agents that are successfully applied in clinical trials [49]. Based on Ehrlich’s concept, therapeutic oligonucleotides represent an advanced class of “smart” drugs that can regulate gene expression patterns involved in the progression of many diseases [50]."

  • Response #5:

We appreciate the comments provided by the reviewer. We have deleted the text as suggested. Please refer to new heading 2.4.

  • Comment#6:

“Line 255: Unclear how SOMAscan relates to aptamers."

  • Response #6:

In line 213 we discuss the SOMAmer strategy. However, following the suggestion of the reviewer, we specified that SOMAscan strategy is based on slow off-rate modified aptamer (SOMAmer). Please refer to line 258.

  • Comment#7:

“Line 473- For the following section (pasted below), I would recommend including the words "when possible" when talking about surgery. Also radiotherapy is used. Often wait and see approach is also used. Finally, some current treatments are prolonging pediatric brain tumor survival."

  • Response #7:

We appreciate the comments provided by the reviewer. We have included the “when possible” wording in the section as suggested. Please refer to lines 500-501.

  • Comment#8:

"Currently, pediatric brain tumor treatments are represented by surgery, the main therapeutic strategy, following by chemotherapy and rarely radiotherapy, due the severe side effects. Molecular mechanisms, behind pediatric brain tumors, remain still limited, but it could represent a new fundamental tool in the therapeutic perspective of these tumors [116].

The current treatments are not prolonging patient survival. There is an urgently need It is needed of for more effective therapeutic approaches. In Table 3, we reported the latest clinical trials for pediatric brain tumors."

  • Response #8:

We have extensively revised heading 4.

This manuscript is a resubmission of an earlier submission. The following is a list of the peer review reports and author responses from that submission.